# Do biomolecular condensates regulate the transcriptional and post-transcriptional responses of plant roots to water deficit?

Coralie Masson[1], Hanzhang Yu[2], Fabrice Bauget[1,3,4], Dominique Gagliardi[2] and Yann Boursiac[1]

[1]Institute for Plant Sciences of Montpellier (IPSiM), Univ Montpellier, CNRS, INRAE, Institut Agro, Montpellier, France; [2]Institut de biologie moléculaire des plantes (IBMP), CNRS, Université de Strasbourg, Strasbourg, France; [3]UMR AGAP Institut, Univ Montpellier, CIRAD, INRAE, Institut Agro, Montpellier, France; [4]CIRAD, UMR AGAP Institut, F-34398 Montpellier, France

## Review

**Keywords:**
biomolecular condensates; post-transcriptional regulation; turgor pressure; transcription; water deficit.

**Corresponding author:**
Yann Boursiac;
Email: yann.boursiac@inrae.fr

**Associate Editor:**
Dr. Yuchen Long

## Abstract

Water deficit at the plant cell level can be assimilated to a reduction in turgor pressure and an increase in osmotic pressure. In a previous work, we showed that the mRNA abundance of some genes displays a quantitative relationship to these physicochemical parameters. Biomolecular condensates have been shown to depend on the physicochemical environment and are known to regulate mRNA fate. In this review, we present recent work about the implication of biomolecular condensates in mRNA regulation of plants under water deficit and question the biophysical origin of their dynamics. Data in the literature suggest that while the perception of mild water deficit may have been overlooked, biomolecular condensates are clear candidates to sense and transduce severe water deficit in plant cells.

## 1. Introduction

Water potential ($\Psi$) is a composite variable indicative of the energy level of water molecules in a system. In plant cells, it is considered to be mainly due to two components: turgor pressure ($P$), which is the hydrostatic pressure exerted by water inside the cells against both the plasma membrane and the load-bearing cell wall, and osmotic potential ($\Pi$), which is related to the interaction between water and other molecules of the cytoplasm (Dainty, 1963). Gradients of $\Psi$, for example between a cell and the apoplast, allow for predicting the net fluxes of water. In the case of a water deficit (WD), external $\Psi$ is lower than that of the cell, thereby provoking an efflux of water, a drop in cell $\Psi$ and a decrease in $P$. In a previous work, we showed that moderate osmotic treatments, as a way to impose WD, provoke quantitative and specific transcriptional responses in the roots of Arabidopsis (Crabos et al., 2023). A cell pressure probe (Hüsken et al., 1978) was used to monitor the impact of an osmotic treatment on the turgor pressure of young root cortical cells. Moreover, the use of a permeating solute (ethylene glycol, EG) also enabled dissociating the effects of a drop in the outer $\Psi$/ external $\Pi$ from those of a decrease in $P$/ inner $\Pi$. Both types of signals triggered their own transcriptional reprogramming: mRNA abundance of some genes was correlated to the (moderate) intensity of the stress applied to the root (external $\Pi$/ outer $\Psi$), while some others were correlated to the cells $P$. These quantitative relationships between mRNA levels and physicochemical parameters are the starting point of this review.

What could be the physicochemical cues inside a plant cell that would end up in quantitatively regulating mRNA abundance? Because a drop in cell $\Psi$ provokes a decrease in $P$ and an increase in $\Pi$, sensing mechanisms for the two components should be looked at. The first type of sensors are proteins able to perceive changes in mechanical cues such as membrane tension and/or its association to the cell wall and, more generally, to cell wall integrity. A few have been identified (Wolf, 2022) and could be related to sensing variations in $P$ upon water deficit. This group includes receptor-like or wall-associated kinases and mechanosensitive channels. The second type of sensors are, up to now, related to chemical cues such as intracellular concentrations in proteins and other solutes or molecular crowding, which we will associate with $\Pi$ for simplicity herein (Parsegian et al., 2000), and which increase when water moves out of the cell due to

a WD. Such sensors include proteins with intrinsically disordered regions or multivalent proteins (Solis-Miranda et al., 2023). These proteins can undergo liquid-liquid phase separation (LLPS) upon physico-chemical variations and form biomolecular condensates.

Biomolecular condensates regulate various aspects of RNA metabolism through the compartmentalization of biochemical reactions or the sequestration of key factors or mRNAs. In the nucleus, biomolecular condensates include the nucleolus, site of ribosome biogenesis and Cajal bodies, whose main functions are the biogenesis, maturation and recycling of small nuclear RNPs (snRNPs) (Hirose et al., 2023; Love et al., 2017). Dicing bodies (D-bodies) are biomolecular condensates in which plant miRNA/miRNA∗ duplexes are produced by dicing complexes containing DICER-LIKE 1 (DCL1), HYPONASTIC LEAVES 1 (HYL) and SERRATE (SE). The phase separation of the multi-functional protein SE is essential for the formation of D-bodies and for the recruitment of DCL1, HYL1 and pri/pre-miRNA to D-bodies (Xie et al., 2021). Nuclear speckles, formed by the binding of transcription factors on regulatory elements on the chromosome, promote efficient transcription by conferring a kinetic advantage. For example, EARLY FLOWERING 3 (ELF3) and FRIGIDA, two transcriptional repressors, form condensates in a temperature-dependent manner, thereby reducing their occupancy at target gene loci and preventing premature flowering (Jung et al., 2020; Zhu et al., 2021). Stress conditions such as viral infection, heat stress and osmotic stress also induce nuclear speckles. In the cytosol, Processing bodies (P-bodies) and stress granules (SGs) are two major classes of biomolecular condensates (reviewed in Decker & Parker, 2012; Hirose et al., 2023). Of note, the appearance, size, number and composition of P-bodies and SGs fluctuate in response to cellular or environmental cues, and they both engage in dynamic interactions and can share some common factors. P-bodies consist of many proteins involved in mRNA turnover and translationally repressed mRNAs. Interestingly, the mRNA content of P-bodies is genetically regulated and, for instance, varies during the cell cycle (Safieddine et al., 2024). Consistent with the fact that the general mRNA decay factors are conserved as P-body core components, P-bodies were originally considered as the main site for mRNA degradation. This initial view has evolved, yet the issue is still debated. RNA degradation can occur with no detectable P-bodies and no abundant degradation intermediates are detected in purified P-bodies (Eulalio et al., 2007; Hubstenberger et al., 2017). However, induced degradation of reporter transcripts leads to their incorporation into the preexisting P-bodies and, although P-bodies are dispensable for mRNA degradation, they offer significantly higher decay rates than the cytoplasm (Blake et al., 2024). The decay of mRNAs targeted to P-bodies is definitely not their sole possible fate because mRNAs sequestered in P-bodies can also return to translation (Bhattacharyya et al., 2006; Brengues et al., 2005). Therefore, the current model postulates that mRNAs in P-bodies are isolated from translation, and can either be destined for degradation or stored for later release. Unlike P-bodies, SGs appear only in stress conditions and are enriched in translation initiation factors (Buchan & Parker, 2009; Kosmacz et al., 2019). The primary function of SGs was believed to be the rapid repression and protection of mRNAs in response to stress. Additionally, a role in sorting transcripts for degradation has been proposed, given that SGs and P-bodies transiently dock with each other and share protein components (Kedersha et al., 2005; Stoecklin & Kedersha, 2013). However, recent evidence has also revealed active translation within SGs (Mateju et al., 2020). From this study, it has been suggested that cycling of transcripts between the cytosol and SGs may be important for stability and translation, which explains why stress-responsive transcripts are found in SGs. Overall, these nuclear and cytoplasmic biomolecular condensates exhibit rapid and reversible dynamics. Although their exact role in RNA metabolism is far from being elucidated (Putnam et al., 2023), biomolecular condensates enable cells to fine-tune gene expression during plant development and in response to stresses.

Such reactivity is particularly important for plants to adapt to fluctuating environmental conditions, including biotic and abiotic stresses such as light, temperature, oxygen and water deficit. Because the topic of biomolecular condensates and water deficit has been reviewed recently (Meneses-Reyes et al., 2024; Romero-Perez et al., 2023), we will specifically focus in this review on:

- Presenting recent and significant work on the role of biomolecular condensates in regulating mRNA abundance under WD,
- Evaluating whether the ranges of cellular physicochemical cues that we identified as transcriptome regulators, $P$ or inner $\Pi$ variations, can regulate the dynamics of condensates in plant cells under water deficit.

## 2. Condensates in the transcriptional and post-transcriptional responses of plants to water deficit

A few landmark studies showing the implication of condensates in regulating mRNA abundance under WD were published over the recent years (Figure 1).

SEUSS (SEU) is a transcriptional adaptor involved in regulating transcription and may play a role in plant growth and development (Bao et al., 2010), as well as in the response to hyperosmotic and salt stress (Wang et al., 2022). Upon osmotic treatments such as sorbitol, KCl or NaCl, SEU rapidly forms nuclear condensates through LLPS (Wang et al., 2022). This condensation is reversible, as SEU-GFP condensates quickly dissolve after stress removal. This mechanism allows SEU to directly sense osmotic changes through molecular crowding and induce its own condensation, which appears to be essential for cellular adaptability to stress. The formation of SEU condensates depends on the presence of intrinsically disordered regions (IDRs), particularly the N-terminal IDR1 domain. Removal of this domain prevents SEU condensation, even though the protein remains functional under normal growth conditions. Indeed, the expression of a truncated SEU variant (SEUΔIDR1) rescues the developmental defects of seu-6 mutant, suggesting that SEU's transcriptional function is not impaired in the absence of stress. However, this mutated variant fails to restore stress tolerance, highlighting the critical role of SEU condensation in stress adaptation. SEU mutant lines exhibit marked hypersensitivity to NaCl and mannitol, as well as an exacerbated response to abscisic acid (ABA), suggesting that SEU functions either upstream or independently of the ABA signaling pathway during osmotic stress. SEU homologs, present in all land plants and characterized by high conservation of an LDB domain and $\alpha$-helices within the N-terminal IDR, form condensates in response to hyperosmotic stress. Analysis of differentially expressed genes (DEGs) in seu-6 mutants reveals that SEU regulates a distinct set of genes under normal and stress conditions; more than half of the DEGs identified under osmotic treatment are stress-specific, indicating that SEU plays a different role depending on environmental conditions. Among these genes, several are directly involved in the osmotic response, and their expression is significantly reduced in seu-6 under sorbitol or NaCl treatment, at both early and late stress stages. Thus, SEU acts as both a

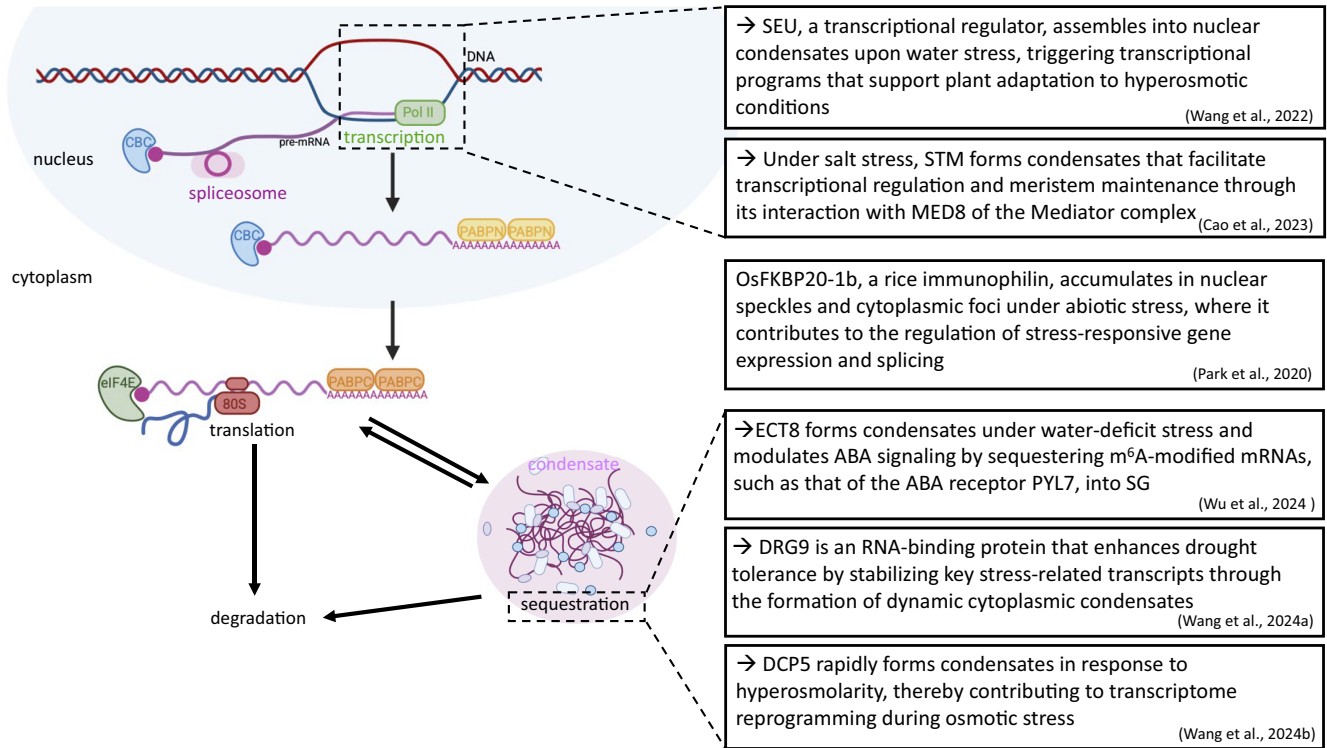

**Figure 1.** Key examples of biomolecular condensates regulating mRNA fate in plants under osmotic stress conditions. *Note*: In addition to their role in mRNA sequestration, biomolecular condensates are involved at various steps of the mRNA life cycle in plants confronted with osmotic stresses. For example, SEU, a transcription regulator, and STM form nuclear condensates under osmotic stress to facilitate transcriptional responses (Cao et al., 2023; Wang et al., 2022). OsFKBP20-1b colocalizes with UBP1b in stress granules and with SR45, a protein involved in splicing (Park et al., 2020). DCP5, meanwhile, exhibits rapid assembly dynamics that are dependent on osmolarity to regulate the transcriptome (Wang et al., 2024b). CBC, cap binding complex; Pol II, RNA polymerase II; PABPN, nuclear poly(A) binding protein; PABPC, cytoplasmic poly(A) binding protein; eIF4E, eukaryotic translation initiation factor 4E.

sensor and a transcriptional regulator of osmotic stress by forming dynamic condensates in response to environmental changes. This condensation ability, which is evolutionarily conserved, is essential for activating transcriptional responses that enable plants to adapt to hyperosmotic stress conditions.

STM (SHOOTMERISTEMLESS) plays a key role in the initiation and maintenance of the shoot apical meristem (SAM) in Arabidopsis (Cao et al., 2023). While stresses such as hyperosmotic stress, hydrogen peroxide ($H_2O_2$), temperature variations and abscisic acid have minimal effects, high concentrations of NaCl ($\geq$ 0.2 M) significantly stimulate the formation of cytoplasmic STM-Venus condensates *in vivo*. The disordered central PrD domain of STM, located in the N-terminal region, promotes this condensation through electrostatic and hydrophobic interactions, which are essential for the organization and regulation of components necessary for meristem activity. In the absence of this domain, as in a truncated STM variant, the ability to form condensates is lost, resulting in a uniform distribution of the protein. Plants with truncated STM exhibit increased sensitivity to salt stress and reduced SAM size. Furthermore, STM interacts with BELL-class TALE homeodomain transcription factors (BEL1-like), a family of plant-specific regulators including ARABIDOPSIS THALIANA HOMEOBOX 1 (ATH1), REPLUMLESS (RPL) and POUNDFOOLISH (PNF) which, through their prion-like (PrD) domains, contribute to the stabilization and regulation of STM-formed condensates. They thus facilitate meristem regulation, particularly of inflorescence meristems, with RPL playing a central role in this process. STM condensates also colocalize with MEDIATOR SUBUNIT 8 (MED8), thereby enhancing its transcriptional activity. In fact, NaCl treatment, which promotes condensation,

increases the expression of STM-regulated genes, while treatment with 1,6-hexanediol, which dissolves the condensates, reduces this activation. This suggests that condensate formation enhances its transcriptional regulatory activity and thus promotes meristem maintenance. Thus, the formation of STM biomolecular condensates plays a crucial role in transcriptional regulation and environmental acclimation, acting as an adjustment mechanism to determine cell fate and response to stress conditions.

OsFKBP20-1b (Park et al., 2020), a protein belonging to the immunophilin family, is expressed at low levels throughout all developmental stages and in all tissues, with some variations during the development of rice plants. In leaves of 4-week-old plants, OsFKBP20-1b localizes to nuclear and cytoplasmic foci, whereas in leaves of 5-week-old plants it redistributes predominantly to the nucleoplasm and cytoplasm, indicating dynamic compartmental mobility during development. OsFKBP20-1b colocalizes with the SG marker OLIGOURIDYLATE BINDING PROTEIN 1b (UBP1b) in the cytoplasm and, additionally, with SR45 (a serine/arginine-rich nuclear ribonucleoprotein and splicing factor) in both nuclear and cytoplasmic speckles. Under salt stress (200 mM NaCl), heat stress (42°C) and other abiotic stresses, the expression of OsFKBP20-1b increases and its subcellular localization becomes more dynamic, with an increase in the number of foci. Consistent with its high reactivity to stress conditions, OsFKBP20-1b plays a key role in regulating stress-responsive genes. Indeed, the loss-of-function mutant shows a significant reduction in the expression of stress-responsive genes such as *LATE EMBRYOGENESIS ABUNDANT 3* (*OsLEA3*) and *RESPONSIVE TO ABA 16* (*OsRAB16*), whereas their transcription levels are tripled in OsFKBP20-1b overexpression lines compared to wild-type plants

in response to ABA. Through its interaction with SR45, OsFKBP20-1b also modulates the alternative splicing of several pre-mRNAs, including those of *OsbZIP60, OsNAC5, OsDREB2B, OsTFIIIA* and *OsRS31*, in response to ABA treatment as well as heat and cold stress, as marked differences in their alternative splicing patterns are observed between the *osfkbp20-1b* mutant and wild-type under these conditions. Moreover, OsFKBP20-1b functions as a molecular chaperone by preventing the degradation of the OsSR45 protein, which is dependent on the 26S proteasome, thereby contributing to the stability and regulation of stress-related proteins. Transgenic seedlings overexpressing OsFKBP20-1b display enhanced tolerance to high salinity and drought compared to wild-type plants, while loss-of-function mutants exhibit growth retardation and increased sensitivity to stress. In summary, RNA processing mediated by OsFKBP20-1b is essential for rice adaptation to stressful environmental conditions, but the role of condensation for its function remains to be determined.

Also in rice, among genes located in a genomic region associated with drought tolerance, *DROUGHT RESISTANCE GENE 9 (DRG9)* stood out as a key candidate due to its strong induction under water stress and drought sensitivity of *drg9* mutants and drought resistance of DRG9 over-expressors (Wang et al., 2024a). The protein is able to bind dsRNA and the 3'-UTR of *OsNCED4* mRNA in particular, which is involved in ABA synthesis. DRG9-YFP forms condensates under mannitol treatment in the root tip through an IDR located in its N-terminus. Mutation within the IDR that disrupts phase separation also abolishes DRG9's ability to stabilize its RNA targets, indicating that LLPS is essential for its biological function. Of note, natural variation of DRG9 in the rice population studied influences mRNA binding capacity rather than condensation properties. Altogether, results in this study position DRG9 as both an mRNA-binding protein and a phase-separation mediator with the capacity to recruit stress-related mRNAs into SG, thereby preserving them from degradation. This mechanism highlights a novel layer of post-transcriptional regulation crucial for plant survival under water-deficit conditions.

Along the same lines, the role of EVOLUTIONARILY CONSERVED C-TERMINAL REGION 8 (ECT8), a protein recognizing the N6-methyladenosine modification ($m^6A$) in mRNAs, was identified for its role in WD signaling in Arabidopsis in 2024 (Wu et al., 2024). In this study, cytoplasmic condensation of ECT8-GFP in SG was observed in root tip cells under various WD-related treatments, including osmotic stresses. This protein contains 3 IDRs, of which IDR #2 is responsible for the condensation properties of ECT8 after polyethylene glycol (PEG) treatment *in vitro* and mannitol *in vivo*. *ect8* mutants show hypersensitivity to ABA and improved fitness upon rewatering after drought. Those phenotypes are not complemented in an *ect8* line expressing the IDR2-deleted ECT8 protein, showing that condensation of ECT8 under stresses is required for its function. Mutations in the $m^6A$ recognition domain also impact the capacity of ECT8 to form condensates. ECT8 appears to recruit $m^6A$ mRNAs into SG, and the mRNA of the ABA receptor PYL7 in particular, reducing its protein level under ABA treatment and providing a negative feedback loop on ABA signaling.

Also in Arabidopsis, the decapping activator/translation repressor DECAPPING 5 (DCP5) was reported to promote condensate formation in response to osmotic stress (Wang, Yang, et al., 2024b). DCP5 was previously characterized as a component of P-bodies, which can associate with SGs in response to heat stress or even promote nuclear condensates controlling flowering in Arabidopsis (Blagojevic et al., 2024; Wang et al., 2023; Xu & Chua, 2009). In

2024, Wang et al. characterized DCP5 as a molecular crowding sensor that directly responds to water deficiency (Wang, Yang, et al., 2024b). Within the IDR of DCP5, a region enriched in highly hydrophobic residues primarily contributes to its condensation ability. While DCP5 homologs in yeast (Scd6) and animals (LSM14A) lack most of this region, it is well conserved in land plants. In Arabidopsis, the *dcp5-1* mutant exhibits increased susceptibility to hyperosmolarity and desiccation, and resistance can only be partially restored by DCP5 lacking this region. In response to molecular crowding induced by cell volume changes under hyperosmotic stresses, DCP5 undergoes LLPS and forms a distinct type of SG, so-called DCP5-enriched osmotic stress granules (DOSGs). DOSGs recruit typical SG components such as mRNAs, translation initiation factors, nucleocytoplasmic transporters and proteins related to the proteasome. Hyperosmotic stress triggers rapid DCP5 condensation within 15 minutes, resulting in altered translation efficiency of thousands of genes. Gene Ontology enrichment analysis revealed a downregulation of growth-promoting genes, along with an upregulation of genes involved in ion transport and stress response. At the transcriptional level, a DCP5-dependent reshaping of the transcriptome can be observed after one hour of hyperosmotic stress, with several hundred genes down-or upregulated. Consistent with the translational regulation, those genes are mainly involved in stress response and growth or developmental processes, respectively. This reprogramming of the transcriptome may be explained by the sequestration of nucleo-cytoplasmic transporters and transcription factors such as BASIC TRANSCRIPTION FACTOR 3 (BTF3), which were found in the proteome of DOSGs. Additionally, nuclear-localized DCP5 has been reported to directly participate in transcriptional regulation: DCP5 co-condensates with SISTER OF FCA (SFF), inhibiting its ability to promote the transcription of target genes. Both DCP5 and SFF contain a prion-like domain (PrLD), which is essential for condensation. Whether DCP5 interacts with other transcriptional regulators in the nucleus and whether this occurs under osmotic stress has not yet been specifically described, which remains a potential direction for further investigation to better understand DCP5's role in stress adaptations.

Taken together, those studies illustrate the potential of biomolecular condensates in sensing WD and in regulating various transcriptional and post-transcriptional steps of gene expression in response to WD. Detailed molecular insights are still required, however. In particular, an added or altered functionality of the condensates compared to the solute phase needs to be assessed to support a model in which condensation can transduce a WD signal (Glauninger et al., 2022; Putnam et al., 2023) -which is already addressed in most of the recent work we reported above. Moreover, a plethora of distinct molecular mechanisms may have been selected to mediate those changes in gene expression and have to be taken into account. These can range from transcriptional regulation to the modulation of mRNA stability or the sequestration of factors or mRNAs away from the translation machinery. In the next section, we propose to add another layer of complexity for future studies, as we discuss why it also seems important to consider other, more biophysical, aspects of water deficit.

## 3. Can physicochemical cues of plant cells under water deficit trigger the formation of condensates?

Biomolecular condensate formation has almost always been associated with an increase in molecular crowding, $\Pi$, or volume change.

However, proteins can also condensate in response to *P* (Cinar et al., 2019), and a drop in *P* is characteristic of the initial phase of WD in plant cells, during which we unraveled quantitative transcriptomic regulations (Crabos et al., 2023). In this section, we therefore explore the ranges of inner *P* or *Π* at which condensate dynamics have been observed in plant cells, and we evaluate whether a condensate-based sensing could occur for *P* or *Π* variations upon WD.

As a technical note, we should stress that both parameters (as well as *Ψ*) are challenging to measure *in vivo* at the cellular level. Direct or indirect measurements of *P* with a cell pressure probe, pico gauges (Hüsken et al., 1978; Knoblauch et al., 2014) or indenters and related techniques (Beauzamy et al., 2016; Zimmermann et al., 2008) are not amenable to all cell types and are of low throughput. Techniques using exogenously applied chemical probes are promising but still require solid foundations for the interpretation of their signals when applied to plant cells (Michels et al., 2020; Ryder et al., 2022) or remain to be effective inside cells (Jain et al., 2021). FRET-based genetically encoded sensors are also very promising, given the increasing knowledge of intra or inter-protein interactions in relation to physicochemical cues and one, Sensor Expressing Disordered Protein 1 (SED1), has been reported for *Π* but has yet to be used *in planta* (Cuevas-Velazquez et al., 2021).

A bibliographic survey for "condensates" and "hydrostatic pressure" retrieved no relevant results in plants, which suggests that the sensitivity of condensates to variations in turgor pressure in plant cells has not been reported yet. Condensates have been shown to be sensitive to hydrostatic pressures *in vitro*, but for variations in the range of a few to hundreds of MPa (Cinar et al., 2019). This exceeds by far what could be physiologically relevant for land plants. Indeed, the root cortical cells of Arabidopsis, barley or maize typically exhibit *P* in the range of 0.18 to 0.70 MPa (Crabos et al., 2023; Ding et al., 2020; Javot et al., 2003; Rygol et al., 1993). The drop in *P* due to an osmotic treatment depends on the nature of the solute and the selectivity of the membrane, but is roughly proportional to the concentration of the solute. For example, we have shown that there is a quasi-linear relationship of the drop in *P* of cortical cells in the young part of the root, in response to 25–150 mM NaCl or sorbitol, or 75–150 g/l PEG8000 (Crabos et al., 2023). Figure 2 (black curve) shows a characteristic turgor pressure change in response to an outer *Ψ* drop. From the literature cited above and other works related to condensates in response to WD (Cao et al., 2023; Chong et al., 2019; Khan et al., 2014; Soma et al., 2020; Wang et al., 2022; Wang et al., 2024a, 2024b), we extracted quantitative indications on the dynamics of molecular condensates and plotted them as a function of *Ψ* in the same figure. It turns out that almost all the data we retrieved fit into a stress intensity zone where root cortical cells would be under plasmolysis. One of the only conditions compatible with a positive *P* was 150mM mannitol treatment on DCP5-GFP expressing plants, which showed a transient formation of condensates (Wang et al., 2024b). Therefore, a variation in *P* in published experiments is unlikely to have driven LLPS and formation of microscopically visible condensates. The absence of a report on condensate formation during milder treatments does not rule out the possibility that it may occur, but this should be tested through dedicated experiments.

The other physicochemical parameter related to water in plant cells is *Π* (or its opposite, the osmotic potential *Ψ_Π*). It is roughly related to the number of solutes within the given volume of a cell. Upon WD, *Π* will be affected in a biphasic way. Because the cell wall is elastic, a change in turgor pressure also impacts cell

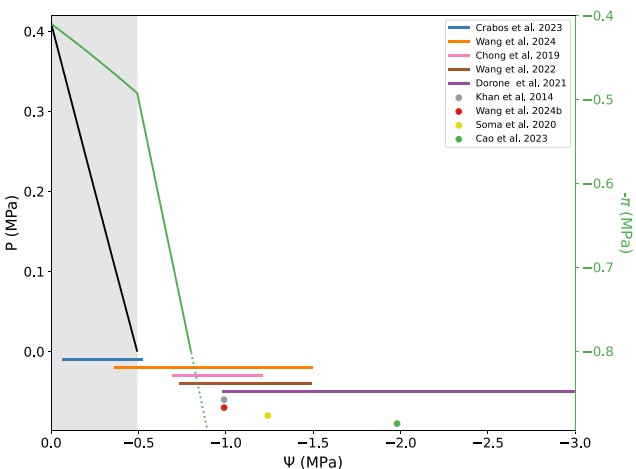

**Figure 2.** Schematic turgor pressure (black line, left axis) and osmotic pressure (green line, right axis) versus water potential of cells during a hyper-osmotic shock. Greyed zone represents the turgid state of the cell, beyond this zone, the cell is plasmolyzed, the turgor pressure is zero and the water potential is equal to $-\Pi$. Horizontal segments and points represent osmotic shock ranges or values found for plant cells in the literature (cf main text). *P* and *Π* were calculated using $\epsilon$ = 2.23MPa and an initial turgor pressure of 0.41MPa. The initial osmotic pressure was set equal to the initial *P*. *Π* was calculated using the mass balance into the cell, where the relative variation of solute concentration, and so the relative osmotic pressure variation, is the inverse of the relative volume variation. *P* is calculated from the elastic modulus definition, and $\Psi = P - \Pi$. Note that some observations were made in the root apical meristem, which could lose turgor at higher osmotic treatments. For more information, see Supplementary Material S1.

volume and therefore will impact *Π*. Considering the definition of the elastic modulus $\epsilon$, $dP = \epsilon \frac{dV}{V}$, *V* being the cell volume and the data obtained with the cell pressure probe on Arabidopsis root cortical cells, $\epsilon$ = 2.23MPa (Javot et al., 2003) and *P*=0.41 MPa (Crabos et al., 2023), we were able to calculate the variation in *Π* together with the variation in *P* while *P* remains positive, i.e. under a mild water deficit treatment (Figure 2, green curve in the gray zone). On average, the maximal volume variation in cortical cells due to turgor loss, thus before plasmolysis, is around 17%, which translates into an increase in *Π* of about 20%. As mentioned earlier for *P*, only one study reported the possible transient formation of DCP5-GFP condensates upon "mild water deficit" (Wang et al., 2024b). With more severe water deficit treatments, *P* is lost and water efflux from the cell will lead to plasmolysis (if solutes can cross the cell wall, such as NaCl or sorbitol) or cytorrhesis (in the case of solutes blocked by the cell wall, such as PEG8000) and, in such case, the increase in *Π* will be directly proportional to the outer osmotic potential (Figure 2, water potentials lower than -0.5MPa). Almost all published works on molecular condensates under WD treatments fall within that range (Figure 2, bottom). Papers cited herein typically have reported increases in *Π* of more than 75 % (see Chong et al., 2019) Figure 2). Hence, the formation of condensates reported thus far is clearly more correlated to *Π* and related parameters than to *P*.

The question of the exact physico-chemical cue underlying our use of *Π* in this review is vast and will not be addressed here. However, we want to point out that several works cited herein have linked molecular crowding sensing to condensation or phase separation (Boyd-Shiwarski et al., 2022; Meneses-Reyes et al., 2024; Wang et al., 2022, 2024b). The relationship between WD and molecular crowding or volume change could be straightforward since an efflux of water from a cell with a selective membrane directly results in a volume reduction and an increase in solute

and protein concentrations. But besides the fact that a biological system is far more complex than an *in vitro* system, such as a dialysis setup, the interpretation of condensation might have to go beyond. Indeed, two driving force categories control LLPS or condensation: entropic forces linked to solute concentrations, and enthalpic forces related to noncovalent interactions, and making the distinction between both is difficult in cells (Romero-Perez et al., 2023). Moreover, Watson and co-authors (Watson et al., 2023) addressed the formation of condensates from the perspective of $\Psi_\Pi$. These authors showed, using two different macromolecules of similar sizes, PEG and Dextran, that PEG (which decreases $\Psi_\Pi$) led to bovine serum albumin (BSA) condensation while dextran (which has a low impact on $\Psi_\Pi$) did not. This result suggests that molecular crowding alone did not provoke LLPS. Furthermore, the condensate state does not always follow the "obvious" molecular crowding way. For instance, at natural expression levels in plant seeds, FLOE1 droplets dissolve during the desiccation process, even though the loss of water increases its intracellular concentration (Dorone et al., 2021; Romero-Perez et al., 2023). Beyond concentration and $\Psi_\Pi$, Watson and colleagues also highlighted the complexity of physical cues acting on LLPS. In particular, using BSA and PEG solutions, they have shown that an increase in temperature could revert PEG-induced BSA condensation. The authors then concluded that molecular crowding could not be the primary condensation factor, but rather the solvent thermodynamics (for instance, temperature, see Figure 4g and h of Watson et al., 2023). Therefore, there is no unique way to understand and anticipate condensate formation in plant cells.

From the works cited above, $\Pi$ rather than $P$ seems thus far to be the physicochemical parameter to work with to better understand and anticipate biomolecular condensate formation. However, solvent thermodynamics influence protein hydration which involves the organization of a few layers of water molecules around macromolecules. This led to the idea of two "kinds" of water: structured water molecules that are influenced by their (in-)direct interactions with macromolecules, and free/bulk water molecules that are "available" for biochemical functions (Romero-Perez et al., 2023; Watson et al., 2023). Hydration of a protein is modified, among others, by its phosphorylation status, which is influenced by osmolarity or temperature changes and implicates condensate formation (Watson et al., 2023). Moreover, it was shown in the animal field that hydrostatic pressure at physiological ranges (80–190kPa) activates WITH NO LYSINE KINASES (WNK, Humphreys et al., 2023). Whether post-translational modifications, mediated through $P$-dependent proteins, can shift the condensation parameters of other proteins to ranges of $P$ observed in plant cells should be investigated. And then, this raises again the interesting possibility that $P$ can be a physical factor acting on the formation of functionally relevant biomolecular condensates in vivo.

## 4. Conclusion

Regulation of mRNAs by condensates under WD has been proven for multiple aspects of mRNA fate but, at the moment, may not explain the quantitative regulations observed for some genes under mild osmotic treatments while $P$ remains positive (Crabos et al., 2023). Indeed, most condensates reported thus far form in response to an increase in $\Pi$ under severe WD rather than a decrease in $P$ under mild WD. This latter aspect has potentially been overlooked. Two "physiological" worlds seem to emerge: before or after plasmolysis, and it is possible that the sensing and signaling

mechanisms towards gene expression are different on each side of this turgor-loss limit. Finally, since the properties of the solvent, PTMs and the combination of molecular actors seem to affect condensation properties, a myriad of combinations of physicochemical conditions and condensates surely remain to be functionally characterized during the lifecycle of plant cells.

**Open peer review.** To view the open peer review materials for this article, please visit http://doi.org/10.1017/qpb.2025.10032.

## Acknowledgements

The authors would like to thank Yunji Huang for the critical reading of the MS. Around 10-15% of the final text benefited from the use of Chat-GPT for text translation and improvement.

**Competing interest.** The authors declare none.

**Author contributions.** CM, FB, HY and YB wrote the first draft of the paper, which was then edited by all authors.

**Funding statement.** FB was supported by Agence Nationale de la Recherche (ANR-22-CE45-0009 EAUDISSECT to YB). CM is supported by the BAP department of INRAE and by Région Occitanie (Feder, Eaudissect). HY and DG are supported by the Interdisciplinary Thematic Institute IMCBio+, as part of the ITI 2021-2028 program of the University of Strasbourg, CNRS and Inserm, and supported by IdEx Unistra (ANR-10-IDEX-0002), EUR (IMCBio ANR-17-EURE-0023) and SFRI STRAT'US project (ANR-20-SFRI-0012) within the framework of France 2030.

**Supplementary material.** The supplementary material for this article can be found at http://doi.org/10.1017/qpb.2025.10032.

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
