## [Reviewer Report]

This is a well-organized and timely review. I have some minor concerns for improving the manuscript.

1. The review would benefit from in-depth discussion on the limitations and challenges within the field. It would be beneficial to dedicate a section to future research directions, possibly outlining specific knowledge gaps and potential experimental strategies to address them.

2.The authors might consider discussing recent technological advancements that could provide further insights into the turgor pressure and osmotic potential in regulating the dynamic of condensates.

3. Figure 1:

Since splicing predominantly occurs co-transcriptionally, it will be more accurate to adjust the position of spliceosome in the figure to reflect this.

The sentence describing OsFKBP20-1b in this figure should be revised to improve clarity.

In the sentence describing ECT8, the chemical notation m⁶A should be formatted using superscript, and a space should be inserted between “into” and “SG”.

4. Use of full names and abbreviations

Please ensure that all abbreviations are defined at their first appearance in the manuscript. In addition, several gene names, such as DCL1, SE, ELF3, and ECT8,are introduced without their full names.

5. In the introduction, the term “FLC-based condensates” requires clarification.

---

## [Reviewer Report]

In this review, the authors address an important and timely question: Do biomolecular condensates (BCs) regulate the transcriptional and post-transcriptional responses of plant roots to water deficit? While the topic is highly relevant—particularly in the context of increasing global water scarcity and the urgent need to understand the underlying molecular mechanisms—the review unfortunately falls short in providing a coherent and comprehensive synthesis of current knowledge.

The manuscript offers only a brief summary of recent studies on BCs and water deficit, but lacks a broader integrative perspective that could clarify how BCs may functionally contribute to plant adaptation under drought conditions. Several key issues limit the impact of the review:

Figure 1 illustrates some structural changes in biomolecular condensates under stress, but it does not propose or support any mechanistic model for how BCs might influence water-deficit responses.

The discussion on water potential (Ψ) and its gradient as a predictor of water fluxes is introduced without a clear or direct connection to the role of BCs, leaving the reader uncertain about the relevance of this section to the central theme.

Much of the review consists of speculative questions rather than a critical synthesis of existing data. As a result, it lacks the structured analysis that would allow the reader to understand what is currently known, what remains unclear, and where the field might go next.

In my view, there remains a significant gap in our understanding of how biomolecular condensates intersect with water potential signaling pathways. Establishing this connection is complex and, at this stage, speculative.

Finally, the review appears to be somewhat misaligned with the scope of the journal. It lacks a quantitative component, which is expected for submissions to this journal.

---

## [Editor Report]

Dr. Boursiac

August 1, 2025

Manuscript number: QPB-2024-0074

Do biomolecular condensates regulate the transcriptional and post-transcriptional responses of plant roots to water deficit?

Dear Dr. Boursiac,

Thank you for submitting your manuscript to Quantitative Plant Biology. I am including the reviewer comments, which I hope you will find useful and constructive. As you will see, they express interest in the review, but they also have several criticisms and suggestions. We encourage you to take all reasonable steps to address their concerns rigorously. Please submit your revised manuscript online and include a point-by-point list of the revisions made to address the reviewers' comments. 

We would like your revision to be submitted by Aug 15, 2025. Should you need additional time, please reach out to me by email.

Thanks again for submitting your work to Quantitative Plant Biology. I look forward to reading your revised manuscript.

Sincerely,

Yuchen Long

yuchen.long@nus.edu.sg

Reviewer #1

This is a well-organized and timely review. I have some minor concerns for improving the manuscript.

1. The review would benefit from in-depth discussion on the limitations and challenges within the field. It would be beneficial to dedicate a section to future research directions, possibly outlining specific knowledge gaps and potential experimental strategies to address them.

2.The authors might consider discussing recent technological advancements that could provide further insights into the turgor pressure and osmotic potential in regulating the dynamic of condensates.

3. Figure 1:

Since splicing predominantly occurs co-transcriptionally, it will be more accurate to adjust the position of spliceosome in the figure to reflect this.

The sentence describing OsFKBP20-1b in this figure should be revised to improve clarity.

In the sentence describing ECT8, the chemical notation m⁶A should be formatted using superscript, and a space should be inserted between “into” and “SG”.

4. Use of full names and abbreviations

Please ensure that all abbreviations are defined at their first appearance in the manuscript. In addition, several gene names, such as DCL1, SE, ELF3, and ECT8,are introduced without their full names.

5. In the introduction, the term “FLC-based condensates” requires clarification.

Reviewer #2

In this review, the authors address an important and timely question: Do biomolecular condensates (BCs) regulate the transcriptional and post-transcriptional responses of plant roots to water deficit? While the topic is highly relevant—particularly in the context of increasing global water scarcity and the urgent need to understand the underlying molecular mechanisms—the review unfortunately falls short in providing a coherent and comprehensive synthesis of current knowledge.

The manuscript offers only a brief summary of recent studies on BCs and water deficit, but lacks a broader integrative perspective that could clarify how BCs may functionally contribute to plant adaptation under drought conditions. Several key issues limit the impact of the review:

Figure 1 illustrates some structural changes in biomolecular condensates under stress, but it does not propose or support any mechanistic model for how BCs might influence water-deficit responses.

The discussion on water potential (Ψ) and its gradient as a predictor of water fluxes is introduced without a clear or direct connection to the role of BCs, leaving the reader uncertain about the relevance of this section to the central theme.

Much of the review consists of speculative questions rather than a critical synthesis of existing data. As a result, it lacks the structured analysis that would allow the reader to understand what is currently known, what remains unclear, and where the field might go next.

In my view, there remains a significant gap in our understanding of how biomolecular condensates intersect with water potential signaling pathways. Establishing this connection is complex and, at this stage, speculative.

Finally, the review appears to be somewhat misaligned with the scope of the journal. It lacks a quantitative component, which is expected for submissions to this journal.